

# Single Event Effect Testing of the PNI RM3100 Magnetometer for Space Applications

Mark B. Moldwin[1], Edward Wilcox[2], Eftyhia Zesta[3], Todd M. Bonalsky[4]

[1]Climate and Space Sciences and Engineering, University of Michigan, Ann Arbor, MI, 48109, USA
[2]NASA Goddard Space Flight Center, Code 561, Greenbelt, MD, 20771, USA
[3]NASA Goddard Space Flight Center, Code 673, Greenbelt, MD, 20771, USA
[4]NASA Goddard Space Flight Center, Code 549, Greenbelt, MD, 20771, USA

*Correspondence to*: Mark B. Moldwin (mmoldwin@umich.edu)

**Abstract.** The results of a destructive single-event effect susceptibility radiation test of the PNI RM3100 magnetometer sensor, specifically the MagI$^2$C ASIC (application specific integrated circuit) on the sensor board are presented. The sensor is a low-resource commercial-of-the-shelf (COTS) magneto-inductive magnetometer. The device was monitored for destructive events and functional interruptions during exposure to a heavy ion beam at the Lawrence Berkeley National Laboratory's 88" Cyclotron. The RM3100 did not experience any destructive single-event effects when irradiated to a total
fluence of 1.4x10$^7$/cm$^2$ at an effective linear energy transfer (LET) of 76.7 MeVcm$^2$/mg while operated at nominal voltage (3.3 V) and elevated temperature (85 °C). When these results are combined with previous total ionizing dose tests showing no failures up to 150 kRad(SI), we conclude that the PNI RM3100 is extremely radiation tolerant and can be used in a variety of space environments.

## 1 Introduction

As part of the University of Michigan's Magnetometer Laboratory's effort to space qualify the PNI RM3100 magnetometer for space applications, we conducted Single Event Effects testing on the commercial-of-the-shelf (COTS) MagI$^2$C application-specific-integrated-circuit (ASIC). The PNI RM3100's performance has an accuracy of about 1.2 nT and noise density of 500 pT/$\sqrt{Hz}$ @ 1 Hz (e.g., Regoli et al, 2017) and is extremely low size (3 x 3 x 2 mm), low mass (5 g), and low-power (5 mW) making it ideal for multi-magnetometer noise cancellation applications (e.g., Sheinker and Moldwin, 2016;
Hoffmann and Moldwin, 2022) that can enable short-boom, boom-less and/or relaxed magnetic cleanliness requirements for magnetometer satellite investigations.

One concern for using COTS electronics for space applications is their long-term reliability and their potential susceptibility to radiation effects, including Single Event Effects (SEE). There are a variety of SEE caused by single energetic particles
(usually heavy ion cosmic rays, trapped radiation belt protons or solar energetic protons). Single Event Upsets (SEUs) are





non-destructive and therefore termed soft errors. They normally appear as transient pulses in logic or support circuitry, or as bitflips in memory cells or registers and can give rise to phantom or false commands (e.g., Moldwin, 2008). In contrast to SEU, there are several types of potentially destructive hard errors that damage or destroy electronics. One type is called Single Event Latchup (SEL) that results in a high operating current, above device specifications, and must be cleared by a power reset (NASA Radiation Effects & Analysis Group, 2021). This paper describes the testing done to study the susceptibility of the PNI RM3100 magnetometer for SEL conducted at the Lawrence Berkeley National Laboratory's 88" Cyclotron (LBNL, 2021).

## 2 Methodology

### 2.1 Devices Under Test

The PNI RM3100 is a printed circuit assembly with three sensor coils, an ASIC, and passive components that provide three-axis magnetic field sensing in a low power and cost assembly (Regoli et al., 2017). It operates on a split 3.3 V analog/digital rail and interfaces to a digital host via standard SPI or I2C serial interfaces.

Throughout this paper, PNI RM3100 is used as common terminology for the device under test (DUT), though only the sole active microelectronic device (the MagI$^2$C ASIC) was irradiated. The RM3100 does not contain any other components susceptible to single-event effects.

The DUTs for this experiment were commercially procured from PNI by the University of Michigan and provided to NASA Goddard Space Flight Center for SEE testing. The plastic package was opened to expose the die for testing, but part markings were not recorded prior to obliteration by combined laser/chemical decapsulation.

Table 1. Part Information

| | |
|---|---|
| Part Number | RM3100 |
| Manufacturer | PNI |
| Device Function | Geomagnetic Sensor, 3-axis |
| Lot Date Code | Unavailable |
| Device Technology | CMOS |
| Quantity Tested | 2 |
| Package | Printed circuit board, with 1 plastic-encapsulated, wire-bonded microcircuit |
| REAG ID# | 21-017 |



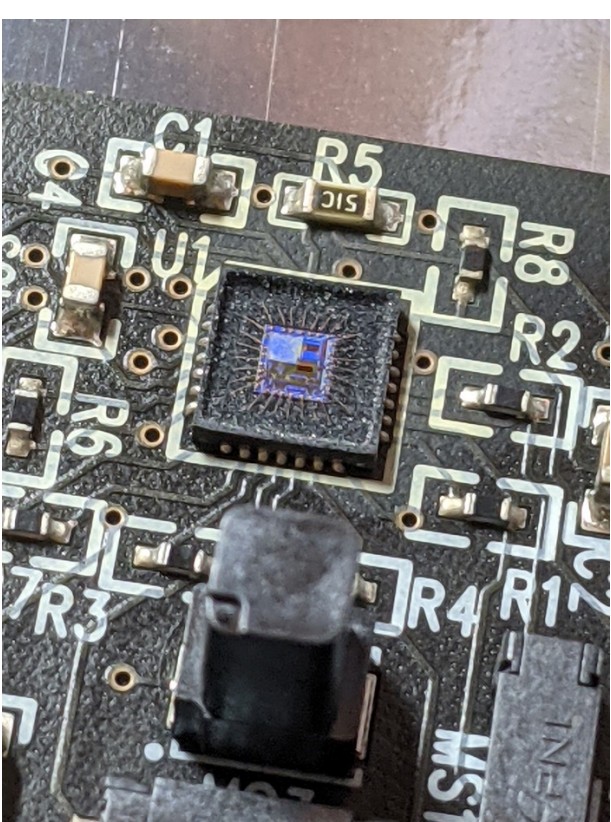


**Fig. 1.** PNI RM3100 ASIC device close-up after decapsulation by SAGE Analytical Labs.

## 2.2 Test Description

The SEE testing was conducted on two decapsulated PNI RM3100 ASICs at the Lawrence Berkeley National Lab 88" Cyclotron, Berkeley Accelerator Space Effects (BASE) Facility. The beam used a 16 MeV/amu tune with a flux varying up

to $1x10^5 cm^{-2}s^{-1}$. Testing was conducted to $1x10^7 cm^{-2}$ at each unique test condition to rule out destructive SEE. Additional tests were performed until Single Event Functional Interrupt (SEFI) were observed (e.g., Koga et al., 1987). SEFI are soft errors that cause the component to reset, or lock-up, but does not require power cycling of the device in contrast to SEL. The test required an effective linear energy transfer (LET) of at least 75 MeVcm$^2$/mg for destructive single-event effect testing. The 16 MeV/amu Xe beam provided a nominal LET of 49.3 MeVcm$^2$/mg in vacuum, and higher effective LETs were

created by irradiating at angles following a $1/cos(\theta)$ rule, until the required 75 MeV cm2/mg was reached. Figure 2 shows the PNI-RM3100 in the beam line prior to the test.



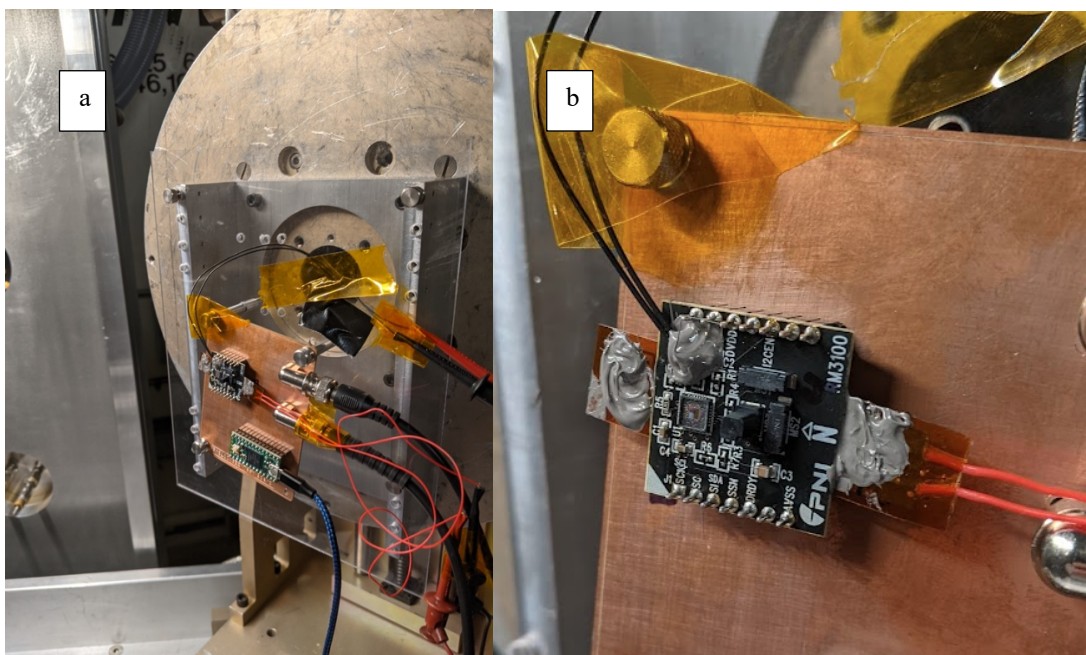

**Fig. 2.** (a) The PNI RM3100 mounted within the beam line for SEE testing. (b) close-up of the decapsulated PNI ASIC with
a polyimide heating strip attached to the back of the PNI RM3100.

Table 2 describes the test conditions that the PNI RM3100 were subject to during the SEE testing.

**Table 2:** *Test Conditions*

| Parameter/Mode | Description |
| --- | --- |
| Test Temperature | Ambient temperature and 85°C package temperature |
| Operating Mode | Static (power only) and dynamic (continuous measurement mode enabled) |
| Power Supply Voltage | 3.3 V nominal and 3.6 V worst-case |
| Parameters of Interest | Power supply current ($I_{CC}$), functional changes in output data |

**2.3 Test Methods**




The RM3100 was controlled by a PJRC Teensy 4.0 ARM™ Cortex-M7 microcontroller (running at 528 MHz), which communicated with the RM3100 via the SPI bus at 1 MHz. The microcontroller received instructions from a host PC running a Python test script. Power supply connections for analog power (AVDD) and digital power (DVDD) were provided independently to isolate any single-event latchup, if observed. For dynamic tests, the RM3100 was configured into

continuous measurement mode, with otherwise default register settings, and readings logged twice per second. For static testing, power was applied to device without reading or writing to any registers.

The primary test was for single-event latchup. Power was supplied at the nominal 3.3 V and the device was operated at 85 °C in static or dynamic mode. Power supply currents were monitored for signs of single-event latchup (a sudden, significant

increase in current only correctable by power cycling).

Characterizing soft errors was a secondary objective of the test. Magnetic readings inside a cyclotron facility are inherently noisy and no attempt was made to calibrate or reference the values to a known standard. Instead, data were monitored for large changes. These errors are collectively counted as single-event functional interrupts (SEFI), and signatures included

sudden data offsets, possible changes in measurement range, frozen data from one or more channels, and lack of communication. They are most likely caused by upsets to the internal control registers, but no attempt was made to readback or automatically correct register values during testing.

## 2.4 Test Results

The RM3100 did not experience any single-event effects when irradiated to a total fluence of $1.4 \times 10^7/\text{cm}^2$ at an effective

linear energy transfer (LET) of 76.7 MeVcm$^2$/mg while operated at nominal voltage (3.3 V) and elevated temperature (85°C) for the durations of the tests. Table 3 shows the test conditions.

**Table 3**. Data Test Run Log

| Facility Setup | | | Test Configuration | | | Run Data | | | Results |
|---|---|---|---|---|---|---|---|---|---|
| Ion | LET | LETeff (MeVcm2/mg) | Voltage (AVDD, DVDD) | Function | Temp (ºC) | Run time (s) | Eff Fluence (/cm2) | Avg Flux (/cm2/sec) | # dSEE |
| Xe | 49.3 | 76.7 | 3.3 | Power only | 85 | 566.2 | 4.08E+06 | 7.21E+03 | 0 |
| Xe | 49.3 | 76.7 | 3.3 | Power only | 85 | 744.8 | 1.00E+07 | 1.34E+04 | 0 |


Single-event functional interrupts (SEFI) were observed, but rare, and most presented as sudden large changes to the measured values on one or more axes and required a power cycle to restore operation. The Threshold LET (LETth) was demonstrated to be greater than 3.7 MeVcm$^2$/mg. The saturated cross-section appears to be less than 1x10$^{-5}$cm$^2$, but data are limited.


## 3 Discussion and Conclusions

The heavy ion beam test results on the susceptibility of the PNI RM3100 magnetometer to SEE found no single event latchup events for LET >75 MeVcm$^2$/mg at an elevated temperature of 85 ºC. Single event functional interrupts (SEFI) were extremely rare. Combined with previous total ionizing dose (TID) tests at the University of Michigan and the NASA

Goddard Space Flight Center (GSFC Radiation Effects Facility, 2021) that found no failures up to 150 kRad(SI) (Regoli et al., 2020), the PNI RM3100 is appropriate for use on missions in a variety of space environments (LEO polar, MEO, HEO, GEO, and deep space).

In addition to TID and SEE testing, the University of Michigan's Magnetic Laboratory is conducting a full range of thermal

and thermal-vacuum testing on the PNI RM3100 exploring both the survival and operation temperature limits and the results will be published in the future to enable the broad use of a COTS magnetometer for both CubeSat and NASA Class C space missions. Currently the PNI RM3100 have been selected for flight on NASA's Artemis Lunar Gateway Heliophysics Environmental and Radiation Measurement Experiment Suite
(HERMES) platform as part of the Noisy Environment Magnetometer in a Small Integrated System (NEMISIS)

magnetometer, and NASA's Heliophysics Flight Opportunities for Research and Technology (H-FORT) Ionospheric Composition and Velocity Experiment (ICOVEX) Satellite. Gateway is scheduled for launch no earlier than 2024, while ICOVEX is scheduled for launch mid-2025.

*Acknowledgements.* This work was supported by NASA H-TIDES (80NSSC18K1240), NASA ICEE2 (80NSSC19K0608)

and NASA HERMES (80GSFC20C0075) grants.

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
