# Peer review of "Single Event Effect Testing of the PNI RM3100 Magnetometer for Space Applications"

_Geoscientific Instrumentation, Methods and Data Systems, 2022_

## Author Response (AR1)

**Referee 1**

Referee Question: "Going forward is there a middle ground for such testing of the PNI RM3100, that is not destructive, but could still be beneficial in increasing the space-worthiness/reliability of this device?"

The standard practice and guidance is that "Microcircuits under test must be delidded [decapsulated]." This is to ensure knowledge of the LET through the circuitry as the cover of the IC could degrade the energy of the beam before reaching the circuitry.

JEDEC, "Test procedure for the management of single-event effects in semiconductor devices from heavy ion irradiation," JESD57A, Nov 2017

https://ieeexplore.ieee.org/document/4638609

"The experimenter should know within reasonable accuracy the LET through the device sensitive volume. The test facility typically reports the initial LET and surface LET as the ion exits the source. However, the experimenter should take care to understand beam degradation through air and other mediums before the sensitive volume. Overburden layers can be significant in some high-density modern ICs. Also, some device types have deep structures that require a long ion range to penetrate the sensitive volume, in order to trigger some destructive effects. So, it is always beneficial to have information on the device dimensions, or be conservative in the beam energy and ion range."

**[Revised Manuscript has the following added (highlighted yellow in revised-Track-Changes pdf).**

(Line 50-51) Decapsulation was performed to ensure reasonable accuracy of LET through the device's sensitive volume.

**Referee 2**

Thanks for the careful reading and review.

*1) The authors note that single event functional interrupts (SEFIs) were observed but rare. Could you quantify how frequently these occur?*

Only very limited SEFI (single-event functional interrupt) testing was done by operating the part in a continuous measurement mode and counting fluence-to-failure, where failure would be a sudden offset in reading, a frozen axis, change of scale, or lack of communications. This test visit did not have the time and the events were not recorded to investigate the small number of events or their cause (likely upset bits in control registers), but the number observed were very small.

*2) The authors note, quite reasonably, that the magnetic readings inside the cyclotron facility are*

*inherently noise and unlikely to be meaningful. Was any attempt made to assess if there was any residual damage from the SEFI events from the testing? For example, did the two tested units perform within manufacturer specifications after testing?*

No performance testing was done on the chips after their decapsulation, only functional testing prior, during and after the beam SEE testing.

**[Revised Manuscript has the following added (highlighted yellow in revised-Track-Changes pdf).**

(Line 103-104)  SEFI events were not recorded as the purpose of the test was to screen for destructive events therefore a SEFI rate cannot be quantified but would be very low.